# Expression of Influenza M2e-NP Recombinant Fusion Protein in *Escherichia coli* BL21 (DE3) and Its Binding to Antibodies

**DOI:** 10.3390/vaccines10122066

**Published:** 2022-12-01

**Authors:** Mei Peng Tan, Noorjahan Banu Mohamed Alitheen, Wen Siang Tan, Wei Boon Yap

**Affiliations:** 1Department of Cell and Molecular Biology, Faculty of Biotechnology and Biomolecular Sciences, Universiti Putra Malaysia, Serdang 43400, Malaysia; 2Department of Microbiology, Faculty of Biotechnology and Biomolecular Sciences, Universiti Putra Malaysia, Serdang 43400, Malaysia; 3Center for Toxicology and Health Risk Studies, Faculty of Health Sciences, Universiti Kebangsaan Malaysia, Jalan Raja Muda Abdul Aziz, Kuala Lumpur 50300, Malaysia; 4Biomedical Science Program, Faculty of Health Sciences, Universiti Kebangsaan Malaysia, Jalan Raja Muda Abdul Aziz, Kuala Lumpur 50300, Malaysia

**Keywords:** influenza A, matrix protein 2 ectodomain (M2e), nucleoprotein (NP), universal vaccine, antigenicity

## Abstract

The current influenza vaccines only confer protection against the circulating influenza subtypes, therefore universal vaccines are needed to prevent upcoming influenza outbreaks caused by emerging influenza subtypes. The extracellular domain of influenza A M2 protein (M2e) is highly conserved among different subtypes of influenza A viruses, and it is able to elicit protective immunity against the viruses. The influenza nucleoprotein (NP) was used to display the M2e in this study due to its promising T-cell response and adjuvanticity. The *M2e* gene was fused to the 5′-end of the *NP* gene and then cloned into pRSET B vector. The DNA sequencing analysis revealed six point mutations in the *M2e-NP* fusion gene, including one mutation in the M2e peptide and five mutations in the NP. The mutations were reverted using PCR site-directed mutagenesis. The recombinant plasmids (pRSET B-M2e-NP and pRSET B-mM2e-NP) were introduced into *Escherichia coli* (*E. coli*) BL21 (DE3) for protein expression. The mutated and non-mutated proteins were subsequently expressed and named mM2e-NP and M2e-NP, respectively. The expression of mM2e-NP and M2e-NP was not affected by the mutations. The binding of anti-M2e antibody to the purified native mM2e-NP and M2e-NP also remained active. However, when the anti-NP antibody was tested, the signal produced by mM2e-NP was very weak. The results implied that the amino acid changes in the NP had adversely impacted on the conformation of mM2e-NP and subsequently affected the antibody binding. In light of the remarkable antibody binding to the M2e-NP fusion protein, this study highly recommends the potential of M2e-NP as a universal influenza vaccine candidate.

## 1. Introduction

Influenza A virus (IAV) infects various species of birds and mammals. It also causes high morbidity and mortality rates in humans during seasonal influenza epidemics and flu pandemics. Each year, the seasonal influenza epidemic causes 3–5 million human infections with more than 250,000 deaths recorded worldwide. Meanwhile, in the past flu pandemics, the virus infections were highly contagious and, therefore, caused reasonably high mortality rate in humans [1]. 

IAV infects the upper and lower respiratory epithelia via interactions between the viral hemagglutinin (HA) and the cellular sialic acid receptor whereas the viral neuraminidase (NA) is involved in the release of newly synthesized virus particles from infected host cells [2]. Most of the licensed influenza vaccines are formulated based on the HA and NA as the dominant antigens. However, both of the antigens are known to have high mutation rates, which usually occurs through two major mutational events, i.e., antigenic shift and antigenic drift [2]. As a result, the mutations greatly reduce the effectiveness of the current influenza vaccines that target influenza A and B subtypes. Therefore, a universal influenza vaccine that contains highly conserved antigens is urgently needed to provide cross-protections against the majority of influenza virus subtypes.

The influenza matrix 2 (M2) antigen is a selective proton ion channel found on the influenza viral envelope. It controls the pH of the interior of virus particles [3]. Upon activation by the low endosomal pH, the M2 ion channels pump in hydrogen ions, which, in turn, acidify the interior of virus particles. The acidic environment promotes the disintegration of the viral nucleocapsid and then releases the viral RNA genome. In addition, the M2 also aids in the maturation of HA by balancing the pH between the trans-Golgi network and cytoplasm [4,5]. The ectodomain of M2 protein, i.e., M2e, has been a popular epitope for the development of universal influenza vaccines, as it undergoes mutations less frequently than the other surface glycoproteins, hence, it is highly conserved among the influenza subtypes [6]. Some studies showed that M2e alone was able to provide partial protection to challenged animals [7,8]. The relatively less-pronounced protection was believed to be due to the small size of M2e. Nonetheless, the M2e-based vaccine was still able to stimulate the production of anti-M2e antibodies that function to promote antibody-dependent cytotoxicity of infected cells, enhance phagocytosis of virus particles, and prevent the release of virus progeny from infected cells [9]. In order to increase the immunogenicity of M2e, it is recommended to couple the M2e with an adjuvant.

Tan et al. [10] proposed the utilization of the influenza nucleoprotein (NP) as an adjuvant and carrier to display the M2e. The proposition was made based on the conserved properties of influenza NP and its ability to initiate T cell responses to fight influenza virus infections [11]. The NP subunits assemble to form virus-like particles (VLPs) [11,12] with the amino terminus extending outward from the VLPs [11]. In view of this, it is believed that by fusing the M2e to the N-terminus of NP, the fusion protein would then oligomerize into VLPs that carry multiple copies of protruding M2e. This not only increases the immunogenicity of the M2e antigen, it also confers the function of NP in activating T-cell responses to the M2e. All in all, the heightened immunogenicity enables better antigen presentation, which, in turn, promotes the activation of humoral, cellular and memory immunity against influenza virus infections.

In this study, the fusion gene of influenza M2e-NP was inserted into pRSET B expression vector and the resultant plasmid was sequenced. A number of mutations were detected and then corrected using PCR site-directed mutagenesis. Both non-mutated (M2e-NP) and mutated M2e-NP (mM2e-NP) proteins were expressed in *Escherichia coli* (*E. coli*) BL21 (DE3). The expression levels and antigenicity of the fusion proteins were validated using western blotting and ELISA, respectively. The study findings indicated that the antigenicity of mM2e-NP was affected by the amino acid changes, although the protein expression level was almost the same as that of M2e-NP. It is, therefore, important to validate the gene sequence of the target antigen and its antigenicity in order to ensure high vaccine efficacy.

## 2. Materials and Methods

### 2.1. PCR Amplification

The viral genome of avian influenza virus (AIV) strain H5N1 (A/chicken/Malaysia/5858/2004(H5N1)) was extracted using Trizol (Invitrogen, Waltham, MA, USA), and 30 ng of the isolated total RNA was converted to first-strand cDNA using random primers (Promega, Madison, WI, USA) and M-MLV reverse transcriptase (Promega, USA). The M2e and NP genes were then amplified from the cDNA using the GoTaq HotStart polymerase (Promega, USA) and specific primers harbouring the BamHI and HindIII restriction sites (single underlined sequences in Table 1). The bold and underlined nucleotide sequences encode for a glycine-serine linker that connects the *M2e* gene sequence to the 5′-terminus of the *NP* gene (Table 1).

The 50 μL PCR reaction mixture contained 1.5 mM MgCl2, 0.5 mΜ mixed dNTPs, 0.5 μM of each specific primer, 2.5 U Taq DNA polymerase, and 5 µL of cDNA. The PCR thermal cycles (BioRad, Hercules, CA, USA) are as follows: 94 °C for 2 min; 35 cycles of 94 °C for 1 min, gradient temperatures: 55, 58.9, 62.5, 67.2 and 71 °C for 1 min, 72 °C for 1 min; and a final extension at 72 °C for 10 min.

The amplified *M2e* and *NP* genes were fused using DyNAzyme EXT DNA polymerase (Fisher Scientific, Hampton, NH, USA), the M2e forward and NP reverse primers (BioBasic, Markham, Canada) (Table 1). The fusion PCR reaction mixture was prepared as aforementioned; both of the M2e and NP PCR products (50 ng each) served as the concurrent templates. The thermal cycles were set as previously described. The resulting PCR product was separated on a 1% (*w*/*v*) agarose gel and visualized on Blue Light Transilluminators (BenchTop, St. Louis, MO, USA).

### 2.2. Construction of pRSET B-M2e-NP Expression Plasmid

The *M2e-NP* fusion gene (1617 bp) was cloned into yT&A cloning vector (Yeastern Biotech, Taipei, Taiwan) and multiplied in E. coli strain TOP 10. The yT&A-M2e-NP plasmid and pRSET B vector were digested with BamHI (Promega, USA) and HindIII (Promega, USA). The digested products were ligated at 1:3 molar ratio (pRSET B: M2e-NP) using T4 DNA ligase (Fisher Scientific, USA). The ligation mixture (5 μL) was added into 100 μL of E. coli strain Top 10, and then incubated at 45 °C for 90 s. About 100 μL of Luria Bertani (LB) broth (Thermofisher, Waltham, MA, USA) was added into the mixture, and incubated at 37 °C on a shaker for 45 min. On LB bacterial agar (Thermofisher, USA), 100 μL of culture was plated using a spreader. The plate was incubated overnight at 37 °C. Bacterial colonies were selected randomly and screened by using PCR. pRSET B-M2e-NP (Figure 1) was isolated and sequenced to confirm the presence of M2e-NP fusion gene and its nucleotide sequence.

### 2.3. Site-Directed Mutagenesis PCR

The DNA sequencing and BLAST results indicated six point mutations that led to amino acid changes in the *M2e-NP* fusion gene (denoted as *mM2e-NP* fusion gene hereafter), in which one of them was in the *M2e* gene, while the other five were found in the *NP* gene. To revert the mutations, 6 pairs of primers (Table 2) were used to perform PCR site-directed mutagenesis. The PCR reaction was prepared as previously described, whilst the thermal cycling steps were carried out as aforementioned. A total of six PCR fragments were generated and subsequently fused using fusion PCR. The resulted fusion gene, *M2e-NP,* was cloned into pRSET B vector, and then introduced into *E. coli* BL21 (DE3) for protein expression.

The single underlined sequences denote the mutated codons, while the double underlined sequences denote the *Bam*HI and *Hind*III restriction sites.

### 2.4. Optimization of Protein Expression (M2e-NP and mM2e-NP Proteins)

Recombinant plasmids encoding M2e-NP and mM2e-NP fusion proteins were introduced into *E. coli* strain BL21 (DE3) for protein expression. Bacterial colonies were inoculated into 10 mL LB broth containing 100 μg/mL ampicillin, and incubated at 37 °C on a shaking incubator for 16 h. Then, 3 mL of the overnight culture was transferred into 150 mL LB broth containing 100 μg/mL ampicillin, and incubated at 37 °C on a shaking incubator (200 rpm) until OD600 reached 0.4. Isopropylthio-β-galactoside (IPTG; 0.5 mM or 1 mM) was added into the bacterial culture. In terms of the optimal expression temperature and period, the protein expression was carried out at 25 °C or 37 °C for a period of 14 h. The counting of the 14-h period began after the addition of IPTG. A total of 10 mL of culture was collected hourly. The cells were spun at 5000 rpm (4000× *g*) for 5 min, and the cell pellet was stored at −80 °C for further analyses. The protein yields were compared to that of the control and expressed as fold of protein expression. The fold-of-protein-expression-versus-incubation-hour curves were plotted using GraphPad Prism version 9.0.

### 2.5. Purification of Recombinant Proteins Using Affinity Column

The bacteria pellet was resuspended with the native binding buffer (ProBond™ Purification System, Thermo Fisher Scientific, USA) containing 1 mg/mL lysozyme to a final biomass concentration of 5% (*w*/*v*). After incubation on ice for 30 min, the suspension was sonicated on ice for six cycles of 10-s bursts at high intensity with 10-s cooling period between each burst. The lysate was centrifuged at 8000 rpm (10,000× *g*) for 10 min. The supernatant containing the soluble protein fraction was subjected to the affinity protein purification using the ProBond resin and column. The ProBond™ resin (2 mL) was added into in a 10-mL purification column, washed with distilled water, and equilibrated with the native binding buffer. Then, the supernatant was added into the column and incubated on a shaker for an hour to allow protein binding. The resin was let to settle by force of gravity and the supernatant was aspirated. The column was washed four times with 8 mL of native wash buffer. Lastly, the protein was eluted with 4 mL of native elution buffer (0.5 mL per fraction). The purified proteins were analyzed with SDS-PAGE and western blotting.

### 2.6. Sodium Dodecyl-Sulfate Polyacrylamide Gel Electrophoresis (SDS-PAGE) and Western Blotting

In the SDS-PAGE, 15% (*w*/*v*) resolving gel was stacked with 5% (*w*/*v*) stacking gel. In sample preparation, 35 μL of protein sample was mixed with 7 μL of 6× Laemmli sample buffer (62.5 mM Tris, 25% (*v*/*v*) glycerol, 0.01 (*w*/*v*) bromophenol blue, 2% (*w*/*v*) SDS, 5% (*v*/*v*) β-mercaptoethanol) and heated at 100 °C for 5 min. The sample was then centrifuged at 10,000 rpm (14,000× *g*) for 1 min. About 10 μL of sample was loaded into a well and the electrophoresis was performed at 100 V for 10 min, followed by 150 V for 90 min. The gel was subsequently subjected to Coomassie blue staining or western blotting. The western blotting was carried out using a tank blotting system. The filter papers and nitrocellulose (NC) membrane were immersed in the transfer buffer. The gel was placed between the NC membrane and filter papers and clamped on a holder. The protein transfer was performed at 100 V for 95 min. After that, the membrane was rinsed with TBS-T buffer (50 mM Tris–HCl, pH 7.6; 150 mM NaCl; 0.1% Tween 20), followed by blocking with skimmed milk [10% (*w*/*v*)] for 1 h in room conditions. The membrane was washed with Tris-buffered saline-Triton X-100 (TBS-T) buffer 3 times, 5 min each. The membrane was incubated overnight with the mouse anti-His antibody (1:2500 in the blocking buffer) at 4 °C. The membrane was washed with TBS-T buffer 3 times, 5 min each. Later, the membrane was incubated with HRP-conjugated goat anti-mouse IgG (1:5000 in the blocking buffer; Fine Biotech, China) at room temperature for 1 h. After that, the membrane was washed again with TBS-T buffer 3 times, 5 min each. The protein band was developed using the enhanced chemiluminescence (ECL) substrate, and the membrane was viewed by using the GelDoc Imaging System.

### 2.7. Enzyme-Linked Immunosorbent Assay (ELISA)

The concentration of the purified M2e-NP and mM2e-NP proteins was determined by Pierce BCA (bicinchoninic acid) Protein Assay (Thermofisher, USA). The target protein (0–1500 ng) was coated on an ELISA plate overnight at 4 °C. Next, the protein solution was decanted and the wells were washed 3 times with TBS-T buffer. Later, the wells were blocked with 1% (*v*/*v*) blocking solution at room temperature for 2 h. The solution was removed and the wells were washed 3 times with TBS-T. Then, 150 μL of primary antibody [mouse monoclonal anti-M2e (1:2500; Abcam, Cambridge, UK) or mouse monoclonal anti-NP (1:3000; Abcam, UK)] were added into the wells, and the plate was incubated on a shaker at room temperature for 1 h. After an hour, the antibody solution was discarded, the wells were washed as previously described. About 150 µL of HRP-conjugated goat anti-mouse IgG was added into the wells and the plate was incubated on a shaker at room temperature for 1 h. Upon the incubation, the wells were emptied, and washed 3 times with TBS-T. For color development, 50 µL of 3,3′,5,5′-tetramethylbenzidine (TMB) substrate was added into each well, and the enzymatic reaction took place at room temperature for 15 min. The plate was read at 605 nm. The OD readings were normalized using the blank controls, and the curves were plotted.

## 3. Results

### 3.1. Amplification and Fusion of M2e and NP Genes

The *M2e* and *NP* genes were amplified from the first-strand cDNA using specific primers. A gradient PCR was carried out to determine the optimal annealing temperature for the reaction. The annealing temperature range of choice was 55 to 71 °C. Figure 2 shows that the *NP* gene (1.5 kb) was amplified at 55–67.2 °C, however, the yield started to decrease when the annealing temperature was increased to 67.2, and finally vanished when the annealing temperature was set to 71 °C (Figure 2, left panel, lanes 4 and 5). The *M2e* gene (132 bp), on the other hand, was only amplified at 71 °C (Figure 2, right panel, lane 5). Primer-dimers were observed at the bottom of the gel when the reaction was performed at 55–67.2 °C. In order to prepare sufficient amplification products for the following fusion PCR, the annealing temperatures, 55 and 71 °C, were set to amplify the *NP* and *M2e* genes, respectively.

### 3.2. Synthesis of M2e-NP Fusion Gene

The *M2e* gene was fused to the 5’-end of the *NP* gene using fusion PCR at 55 °C. In the reaction mixture, the *M2e* forward and *NP* reverse primers were used to fuse the amplified genes. In parallel to the calculated molecular size of the fusion gene (*M2e-NP* fusion gene), a DNA band of ~1.6 kb was visualized on a UV transilluminator. In order to precisely justify the presence of *M2e-NP* fusion gene, three control reactions were prepared, i.e., a positive control for PCR (Figure 3, lane 1), and two negative controls in which either one of the templates used in the fusion of *M2e* and *NP* genes was added (Figure 3, lanes 2 and 3). Figure 3 shows the presence of a DNA band of ~1617 bp on the agarose gel, therefore the *M2e* and *NP* genes were successfully fused as the *M2e-NP* fusion gene (lane 4).

### 3.3. Selection of Positive Clones Carrying pRSET B-M2e-NP and DNA Sequencing

*E. coli* strain Top10 cells carrying pRSET B-M2e-NP were screened and selected using PCR. The positive clones were cultured and subjected to plasmid DNA extraction. The isolated DNA was then sequenced in order to identify the nucleotide sequence of the *M2e-NP* fusion gene.

The sequencing results indicated that the nucleotide sequence of *M2e-NP* was 98% similar to that of influenza A virus H5N1 ((A/chicken/Malaysia/5858/2004 (H5N1)). In the *M2e-NP* fusion gene, six point mutations leading to amino acid changes in the M2e-NP protein were detected, in which a leucine residue was replaced by an isoleucine at location 25 in the M2e whilst the other five replacements were found in the NP (Table 3): D_353_N, R_361_Q, T_407_A, A_445_T and K_480_R.

The mutations were corrected to that of A/chicken/Malaysia/5858/2004 (H5N1) using site-directed mutagenesis PCR. Both of the non-mutated (M2e-NP) and mutated M2e-NP (mM2e-NP) were expressed and purified; the effect of amino acid changes on the protein antigenicity was determined using western blotting and ELISA.

### 3.4. Optimization of M2e-NP Protein Expression

The M2e-NP and mM2e-NP proteins were expressed by *E. coli* strain BL21 (DE3). Three critical protein expression parameters, i.e., the incubation temperature, IPTG concentration and expression period were optimized in order to maximize the protein yield and solubility.

The cell lysate (5% biomass, *w*/*v*) was first separated on a polyacrylamide gel, and the proteins were then detected using the mouse anti-His monoclonal antibody (Elabscience, Wuhan, China) in the western blotting (Table 4 and Figure 4). The empty *E. coli* BL21 (DE3) was adopted as the negative control for protein expression. Comparing the expression of M2e-NP and mM2e-NP at 25 °C and 37 °C, they were better expressed at 37 °C. At 25 °C, regardless of the concentrations of IPTG used in the induction of protein expression, the protein yields were relatively lower, with the majority of M2e-NP and mM2e-NP remaining in the soluble fractions (Table 4, and Figure 4a,b). Due to the negligible insoluble protein yields of M2e-NP induced with 0.5 and 1.0 mM IPTG, the protein bands were barely detected by the anti-His antibody (Table 4); as a result, the quantitative comparison to that of control was less likely to perform in this study (Figure 4b). Besides, mM2e-NP also seemed to be better expressed than M2e-NP at 25 °C (Figure 4a).

As observed in Table 4, Figure 4c,d, due to a more rapid growth at 37 °C, *E. coli* BL21 (DE3) was able to constantly produce M2e-NP and mM2e-NP for a duration of 14 h. The protein yields peaked after 4–7 h of induction, then started to decrease after 8–14 h post-induction. In bacterial expression systems, native proteins are usually harvested from the soluble fraction, whereas insoluble proteins usually remain in the aggregate fraction [13,14]. In vaccine development, native proteins are preferred to insoluble aggregates as they are more structurally and antigenically similar to the antigens found on pathogens, and therefore are able to trigger effective immunity against the target pathogens [10]. In this study, the solubility of the proteins (expressed at 37 °C, Table 4, Figure 4c,d) was affected by the concentration of IPTG. The soluble proteins were more readily produced with 0.5 mM IPTG than 1.0 mM IPTG (Figure 4c). At both IPTG concentrations, insoluble proteins were found in the pelleted aggregates (Figure 4d). At 37 °C, the expression levels of M2e-NP and mM2e-NP were not affected by the amino acid replacements. However, it does not rule out the effect of the mutations on the protein antigenicity. In order to produce M2e-NP and mM2e-NP optimally for affinity protein purification and antigenicity test, the protein induction was performed using 0.5 mM IPTG at 37 °C and the expression period was set to 5 h.

### 3.5. Affinity Purification of M2e-NP and mM2e-NP Proteins

The M2e-NP and mM2e-NP were extracted from the soluble fraction and then purified using the ProBond Affinity Purification column. The proteins were collected in fractions (0.5 mL/fraction) in order to ensure high protein yield and purity. Compared to the crude lysates (lane 2, Figure 5a,b), the purified proteins appeared as single bands on the gels. The migration of the fusion proteins was affected by their relatively high pI (~8.5), which, in turn, influences their overall charges in the resolving gel (pH 8.8). As a result, the protein bands were positioned higher than the estimated molecular size, i.e., 64 kDa. In view of that, to further verify the fusion proteins, and to gauge the influence of the mutations on the antigenicity of the proteins, they were subsequently tested in the western blotting using the mouse monoclonal anti-M2e antibody (Abcam, UK) and the mouse monoclonal anti-NP antibody (Abcam, UK).

In the purified mM2e-NP, some impurities remained and co-eluted with the target protein, even though a stringent washing was adopted (Figure 5b, lanes 4 & 5). In view of that, only fractions 6 and 7 (Figure 5b) were pooled, and subjected to the antigenicity test in order to avoid unspecific antibody binding in the test.

### 3.6. Antigenicity of M2e-NP Protein

The antigenicity of the purified M2e-NP and mM2e-NP proteins was first verified with western blotting using the anti-M2e and anti-NP antibodies. In the western blotting (Figure 6a,b), single protein bands were detected using the antibodies, hence, there was high antibody binding specificity. Nonetheless, the heat denaturation prior to the SDS-PAGE and western blotting linearized the proteins and exposed the epitopes readily to the antibodies, the fusion proteins were therefore equally detectable using both of the antibodies. The effect of the mutations on the antigenicity of M2e-NP and mM2e-NP was hardly discernible in the western blot (Figure 7a,b).

Of note, the impact of the mutations on the antigenicity of the proteins was further verified in the ELISA (Figure 7). Unlike the linearized proteins detected in the western blotting, the purified proteins coated on the 96-well plates were structurally intact; therefore, the binding of antibodies to the native proteins enables the prediction of the protein antigenicity and efficacy in triggering the host immunity.

In order to test the sensitivity of the antibodies to mM2e-NP and M2e-NP, 50–1500 ng of proteins were coated on the wells (Figure 7a). When tested using the anti-M2 antibody (1/2500 dilution), the signals were less obvious with 50 and 100 ng of proteins. The OD readings increased gradually with 500 ng of mM2e-NP (0.3) and M2e-NP (0.2), 1000 ng of mM2e-NP (0.35) and M2e-NP (0.3), and 1500 of ng mM2e-NP and M2e-NP (nearly 0.4 for both of the proteins). The results indicated that the single nucleotide substitution in the M2 protein did not affect the binding of the antibody to M2e-NP.

Meanwhile, when testing M2e-NP with the anti-NP antibody (1/2500), the OD_605_ readings increased (approximately 0.05 to ~0.35) with the ascending protein amounts (Figure 7a). In contrast, the signal was remarkably low (below 0.1) with mM2e-NP as the binding counterpart of the antibody. The results implied that the point mutations in NP had altered the affinity of the antibody to the epitopes available on the exterior of mM2e-NP, hence the reduced antibody binding.

The sensitivity of the antibodies to the proteins was further verified using a series of antibody titers (1/2500, 1/3000, 1/4000, 1/5000, and 1/6000). Since a remarkable difference was observed between 100 and 500 ng proteins (*p* < 0.001) (Figure 7a), the 96-well plates were coated with 100 (Figure 7b) and 500 ng (Figure 7c) proteins separately, and subsequently tested with the series of antibody titers. With 100 ng of proteins, the signals were very low (0.03–0.08) regardless of the antibodies; the signals were almost barely detectable when detecting mM2e-NP with the anti-NP antibody (Figure 7b, 1/3000 to 1/6000 dilutions). The OD_605_ readings were higher when the amount of coated proteins was increased to 500 ng. Greater readings (0.2 and 0.3) were recorded when the anti-M2 antibody was diluted 1/2500. When higher anti-M2 antibody titers were used, the signals decreased drastically, in which OD_605_ ~ 0.1 for 1/3000 and 1/4000 and < 0.1 for 1/5000 and 1/6000).

On the other hand, when 500 ng of M2e-NP and mM2e-NP were incubated with the anti-NP antibody, the antibody seemed to bind better to M2e-NP (between 0.1–0.15) than mM2e-NP (Figure 7c). The OD_605_ signal was still very low (~0.05) for mM2e-NP even at 1/2500 dilution. The readings for mM2e-NP were barely detected when the titers were increased to 1/3000 and higher. In a nutshell, the amino acid changes in the NP had affected the binding of the antibody. The purified M2e-NP still remained antigenic as the active titers of anti-M2e (1/2500) and anti-NP (1/3000) antibodies were in line with that recommended by the manufacturer.

## 4. Discussion

Owing to the constant and continuous mutations in the HA and NA genes of influenza A viruses, the efficacy of the existing HA- and NA-based influenza vaccines are affected, and thereby, are not able to prevent influenza epidemics and seasonal flu occasionally [15]. In the effort to seek effective universal influenza vaccines that are effective against a wider range of influenza subtypes and are not easily affected by mutations in the HA and NA proteins, the potential of highly conserved surface antigens, such as M2e, has been explored [6,16]. M2e is an external peptide found on the M2 channel and is responsible for creating an acidic condition in the endosomes, which, in turn, helps liberate the viral RNA genome from the viral nucleoprotein [17]. In this light, inhibiting M2e with antibodies raised via vaccination can block influenza virus replication in host cells, and thereby, can terminate the spread of viral infection.

However, M2e has been shown to induce relatively lower immunity in hosts, it therefore requires an adjuvant carrier, such as the influenza NP protein, to increase its immunogenicity. In addition, the ability of NP to assemble into virus-like particles (VLP), which mimic the structures and morphologies of native virions adds to its benefits as a carrier [12,18]. Fusing M2e peptide to the N-terminus of NP enables the display of multiple units of M2e on the outside of VLP, which, in turn, increases the immunogenicity of M2e [10]. Furthermore, M2e-NP VLPs that lack the viral genome and infectious ability are safer for vaccine recipients, including those in high-risk groups [19]. A relatively shorter preparation time is needed to prepare VLP vaccines compared to egg-based influenza vaccines [20,21]. Furthermore, the influenza NP is a promising activator of CTL. This prominent CTL-activating property is transferrable to the M2e, hence conferring even greater immunogenicity [22,23,24]. As a result, the immunity raised using M2e-NP fusion protein is able to prevent influenza virus infections [10,25].

Firstly, in order to successfully clone the *M2e* gene at the 5′-end of the *NP* gene, the primer annealing temperature was optimized. This enables the specific primers to attach specifically to the target sequence so that the target genes are amplified [26]. The optimum annealing temperatures for the amplification of the *M2e* and *NP* genes were 71 °C and 55 °C, respectively. The higher annealing temperature required for the amplification of the *M2e* gene was most likely due to the greater G-C content in its coding sequence [27]. To link the M2e to the N-terminus of NP, a glycine-serine linker was adopted. The glycine-serine linker renders greater structural flexibility and increases the solubility of the fusion protein via the formation of hydrogen bonds with water molecules. In addition, the presence of a glycine-serine linker in the fusion protein also improves the protein stability and oligomerization, and its resistance to protease degradation [28,29].

The *M2e-NP* fusion gene was cloned into pRSET B expression vector that enables high protein yield in *E. coli* BL21 (DE3) without deteriorating the conformation and antigenicity of the fusion protein throughout the protein expression period [30]. Prior to the protein expression and antigenicity analyses, the coding sequence of *M2e-NP* fusion gene were analyzed and the results indicated a few nucleotide replacements in the *M2e-NP* fusion gene, including a point mutation and five nucleotide substitutions in the *M2e* and *NA* genes, respectively. Those mutations could lead to low immunization efficacy if they happen to modify the amino acid composition and molecular structure of the fusion protein [31,32]. In order to investigate the impact of the mutations on the antigenicity of the M2e-NP, the point mutations were corrected using PCR site-directed mutagenesis, the expression and antigenicity of the mutated and non-mutated proteins were then analyzed and compared.

When expressing recombinant proteins in bacterial expression systems, the proteins and production processes usually render some physiological impacts to the bacterial hosts [33]. When the protein expression is performed at a relatively high temperature and with a considerable amount of IPTG as an inducer, it expedites the growth of the bacterial host, hence promoting a more rapid protein expression. Owing to the rapid protein production, the protein assembly and folding may not occur constitutively. As a result, the majority of the protein is produced as insoluble aggregates that do not favor the vaccine production [33,34]. To circumvent the aforementioned situations, the protein expression temperature (25 vs. 37 °C) and IPTG concentration (0.5 vs. 1.0 mM) were optimized in this study. The results showed that most of the soluble M2e-NP and mM2e-NP were produced at 37 °C with 0.5 mM IPTG. The yield seemed to peak around 5–6 h post-IPTG induction. The optimal expression period was set at 5 h after considering the time and cost involved in the protein expression. Interestingly, the fusion proteins were not toxic to the bacterial cells as the bacterial biomass did not decrease over the expression period (results not shown). At 25 °C, the protein yield of mM2e-NP was higher than M2e-NP protein. This is likely to be due to biases in the codon use by *E. coli* BL21 (DE3) in the protein translation; however, when the incubation temperature increased, the biases were compensated by the rapid bacterial growth [35]. Following the optimization experiments, M2e-NP and mM2e-NP were produced in *E. coli* BL21 (DE3) with 0.5 mM at 37 °C for 5 h prior to the affinity protein purification, western blotting and ELISA. This was expected to minimize the metabolic burden experienced by the bacterial cells, hence, higher protein production yields [36].

As described earlier, amino acid changes in antigens can cause detrimental effects to the antigenicity of the antigen and its efficacy as a vaccine [37]. In this study, the antigenicity of the purified M2e-NP and mM2e-NP was verified using anti-M2 and anti-NP antibodies. The single nucleotide mutation in the M2e gene did not alter the binding of anti-M2 antibody to the gene product (mM2e-NP). This explains the stability of M2e as a highly conserved peptide and is therefore able to induce protective immunity against various influenza virus subtypes [38]. In addition, the binding of anti-M2e antibody to the native M2e-NP and mM2e-NP proteins in the ELISA also implies the display of M2e peptides on the exterior of VLP, and therefore, is expected to improve the immunogenicity of the M2e-NP as a universal influenza vaccine.

There were five mutational points found within the carboxyl terminus of NP. Unfortunately, some of them (particularly T_407_A and K_480_R) coincided with the immunodominant epitopes of NP, i.e., NP_397–420,_ and NP_463–475_ that are responsible for IAV-specific CD4+ T cell responses [39]. Hypothetically, mutations within antigenic epitopes can modify the protein–protein interactions in the region and subsequently change the protein conformation and antibody binding affinity [40]. This hypothesis is in line with the binding affinity of anti-NP antibody to mM2e-NP in this study. In its linearized form, mM2e-NP was readily detected by the antibody in the western blotting. Nonetheless, when testing the purified native mM2e-NP in the ELISA, the binding of the anti-NP antibody to mM2e-NP occurred very minimally compared to that of M2e-NP. The observation indicates that the antigenic epitopes are located in the C-terminus of NP (residues 353, 361, 407, 445, and 480), and it is, therefore, important to restore the amino acid sequence in that region in order to ensure high antigenicity and immunogenicity of the fusion protein.

According to the ELISA results, the higher the protein amounts, the greater the signals. In this light, it is essential to determine the amount of fusion protein required for inducing sufficient immune response [41,42]. It was shown that at least 500 ng of M2e-NP was required for antibody binding at the recommended titers, i.e., 1/2500 and 1/3000 for anti-M2e and anti-NP, respectively. Surprisingly, the anti-NP antibody was still able to bind to M2e-NP and produce readable signals at higher titers (1/4000–1/6000). This implies that the antibody has a reasonably good binding to the fusion protein. This study underpins the need of taking any nucleotide and amino acid changes in antigens into account prior to making them into vaccines. Unintentional amino acid changes can lead to detrimental conformational changes in a vaccine and reduces its efficacy.

## 5. Conclusions

Production of M2e-NP fusion protein in *E. coli* BL21 (DE3) promises a rapid, less tedious, and cost-effective protein expression system without compromising the binding of anti-His, anti-M2e, and anti-NP antibodies to the fusion protein. In addition, the relatively greater binding of anti-M2e antibody to the fusion proteins in ELISA implies that M2e peptides are displayed readily on the exterior of the M2e-NP VLPs. However, it is important to verify the nucleotide and amino sequences of a vaccine candidate prior to proposing its use as a prophylactic vaccine. As indicated in this study, point mutations in the nucleotide sequence, which, in turn, lead to amino acid changes had detrimentally affected the antigenicity of mM2e-NP. Based on the collected data, it is suggested to further investigate the biochemical and biophysical properties of the M2e-NP VLPs. On top of that, the fusion protein can be subjected to vaccination, immunization, and challenge studies in animal models in order to better adopt the fusion protein as a universal influenza vaccine.

## Figures and Tables

**Figure 1 vaccines-10-02066-f001:**
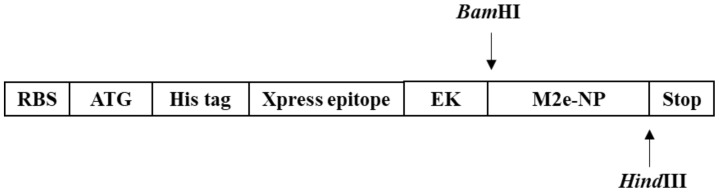
Gene arrangement in pRSET B-M2e-NP plasmid. The fusion gene was flanked by a histidine tag, Xpress epitope and enterokinase (EK) cleavage site on its 5′-end. The *Bam*HI and *Hind*III restriction sites were located at its 5′- and 3′-end, respectively.

**Figure 2 vaccines-10-02066-f002:**
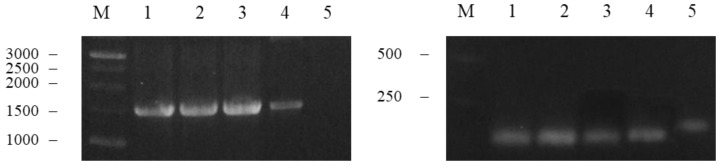
Amplification of the *NP* (**left**) and *M2e* (**right**) genes in a gradient PCR. Lanes M, 1 kb DNA markers; 1, 55 °C; 2, 58.9 °C; 3, 62.5 °C; 4, 67.2 °C and 5, 71 °C. The amplification yield of the *NP* gene (~1.5 kb) decreased with the increasing annealing temperatures whereas the *M2e* gene (132 bp) was only amplified at 71 °C.

**Figure 3 vaccines-10-02066-f003:**
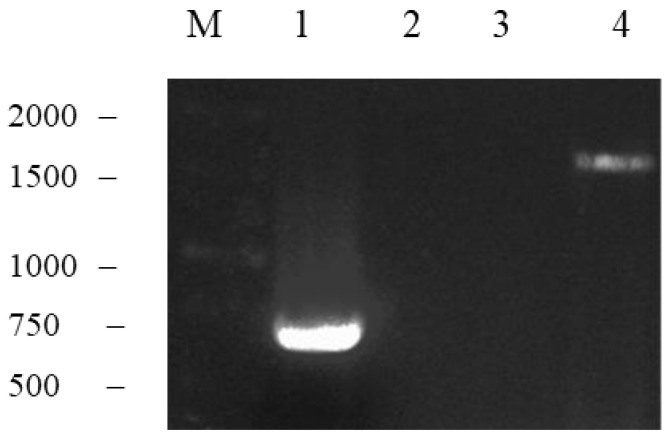
Fusion of *M2e-NP* gene. Lanes M, 1 kb of DNA markers; 1, positive control for the PCR reaction; 2, the PCR reaction containing the previously amplified *M2e* gene as the template (negative control); 3, the PCR reaction containing the previously amplified *NP* gene as the template (negative control); 4, *M2e-NP* fusion gene.

**Figure 4 vaccines-10-02066-f004:**
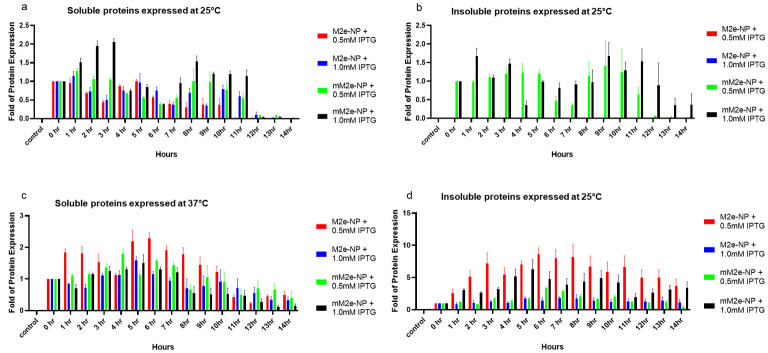
Folds of protein expression compared to the fusion proteins harvested at 0 h (prior to the addition of IPTG). The upper and lower panels represent protein yields at 25 °C and 37 °C, respectively. (**a**) Soluble and (**b**) insoluble M2e-NP and mM2e-NP harvested at 25 °C, and (**c**) soluble and (**d**) insoluble M2e-NP and mM2e-NP harvested at 37 °C. The quantitaion was performed in triplicate (*n* = 3), the data were presented as mean ± SD.

**Figure 5 vaccines-10-02066-f005:**
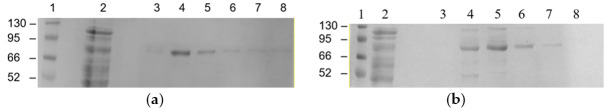
Purification and Coomassie Blue-stained (**a**) M2e-NP and (**b**) mM2e-NP using the ProBond Affinity Purification column. Lanes 1, Blue Ultra Pre-stained protein ladder (Cleaver Scientific, Rugby, UK); 2, crude lysate; 3–8, elution fractions (0.5 mL each).

**Figure 6 vaccines-10-02066-f006:**
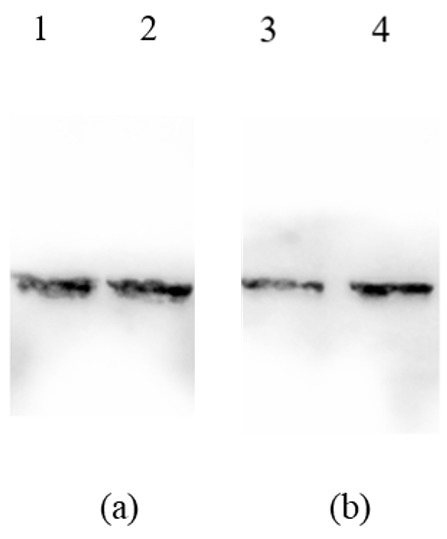
Detection of the purified M2e-NP and mM2e-NP proteins using (**a**) mouse anti-M2e and (**b**) mouse anti-NP antibodies in the western blotting. Lanes 1 & 3, mM2e-NP; 2 & 4, M2e-NP protein.

**Figure 7 vaccines-10-02066-f007:**
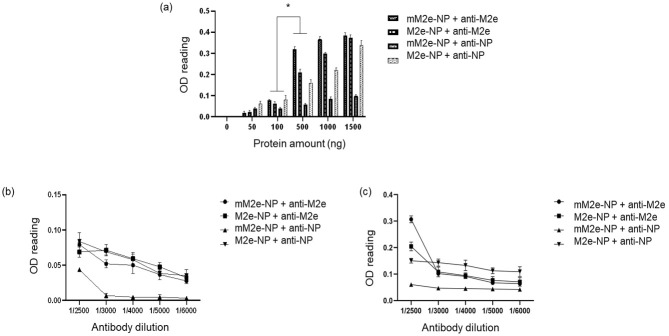
Testing the antigenicity of M2e-NP and mM2e-NP with ELISA. (**a**) The test was performed using 50–1500 ng proteins. The test was also performed using fixed amounts of proteins, i.e., (**b**) 100 ng and (**c**) 500 ng. The experiment was performed in triplicate (*n* = 3), the data were presented as mean ± SD. * indicates *p* < 0.001.

**Table 1 vaccines-10-02066-t001:** Primer sequences used for the amplification of the influenza M2e and NP genes.

Primer	Sequencing
*M2e* forward primer	5′-ATA AAG GAT CCG AGT CTT CTA ACC GAG GTC GAA ACG CCT ACC AGA AAC GAA-3′
*M2e* reverse primer	5′-GGT GCC TTG AGA CGC ACC ACC ACT ACC ACC ATT TGC GGC AAC AGC-3′
*NP* forward primer	5′-GCT GTT GCC GCA AAT GGT GGT AGT GGT GGT GCG TCT CAA GGC ACC-3′
*NP* reverse primer	5′-TCA AGC TTC ATT GTC ATA CTC CTC TGC-3′

**Table 2 vaccines-10-02066-t002:** Primers used in PCR site-directed mutagenesis.

Primer	Sequence
Forward 1F(*M2e* forward primer)	5′-ATA AAG GAT CCG AGT CTT CTA ACC GAG GTC GAA ACG CCT ACC AGA AAC GAA-3′
Reverse 1R	5′-GCC TTG AGA CGC ACC ACC ACT ACC ACC ATT TGC GGC AAC AGC AAT AGG-3
Forward 2F	5′-GCT GTT GCC GCA AAT GGT GGT AGT GGT GGT GCG TCT CAA GGC ACC-3′
Reverse 2R	5′-ACT CTT ATG TGC TGG ATT CTC ATT TGG TCT AAT-3′
Forward 3F	5′-GAG AAT CCA GCA CAT AAG AGT CAA TTA GTG TGG-3′
Reverse 3R	5′-GGA GTC CAT TGC CTC CAT GTT CTC-3′
Forward 4F	5′-GAG AAC ATG GAG GCA ATG GAC TCC-3′
Reverse 4R	5′-TAC CGA GAA AGT GGG CTG AAC GCT-3′
Forward 5F	5′-AGC GTT CAG CCC ACT TTC TCG GTA-3′
Reverse 5R	5′-ACT TTC CAT CAT TCT TAT GAT TTC-3′
Forward 6F	5′-GAA ATC ATA AGA ATG ATG GAA AGT-3′
Reverse 6R(*NP* reverse primer)	5′-TCA AGC TTC ATT GTC ATA CTC CTC TGC-3′

**Table 3 vaccines-10-02066-t003:** Nucleotide and amino acid changes found in the *M2e-NP* fusion gene.

Gene	Nucleotide Change(5′-3′)	Amino Acid Change
*M2e*	…CTT… …ATT…	Leucine (L_25_) → Isoluecine (I_25_)
*NP*	…GAT… …AAT…	Aspartic acid (D_353_) → Asparagine (N_353_)
	…CGA… …CAA…	Arginine (R_361_) → Glutamine (Q_361_)
	…ACA… …GCA…	Threonine (T_407_) → Alanine (A_407_)
	…GCT… …ACT…	Alanine (A_445_) → Threonine (T_445_)
	…AAA… …AGA…	Lysine (K_480_) → Arginine (R_480_)

The numbers denote the locations of the amino acids in the M2e-NP protein.

**Table 4 vaccines-10-02066-t004:** Optimization of M2e-NP and mM2e-NP protein expression.

Expression Temperature ( °C)	IPTG Concentration (mM)	Protein	Type of Lysate	Western Blotting Results
25	0.5	mM2e-NP	Supernatant/Soluble	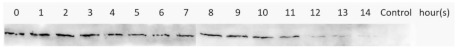
Pellet/Insoluble	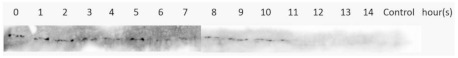
M2e-NP	Supernatant/Soluble	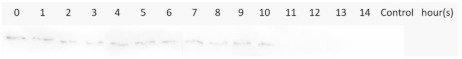
Pellet/Insoluble	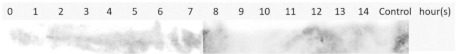
1.0	mM2e-NP	Supernatant/Soluble	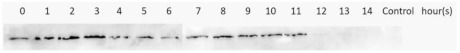
Pellet/Insoluble	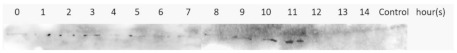
M2e-NP	Supernatant/Soluble	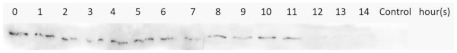
Pellet/Insoluble	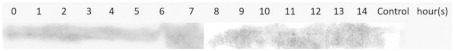
37	0.5	mM2e-NP	Supernatant/Soluble	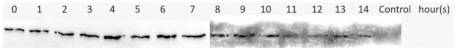
Pellet/Insoluble	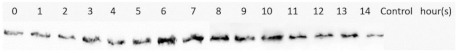
M2e-NP	Supernatant/Soluble	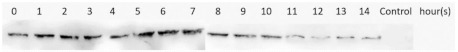
Pellet/Insoluble	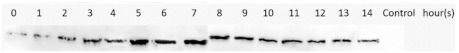
1.0	mM2e-NP	Supernatant/Soluble	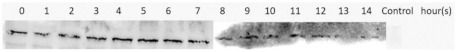
Pellet/Insoluble	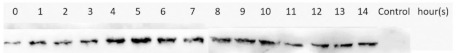
M2e-NP	Supernatant/Soluble	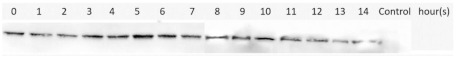
Pellet/Insoluble	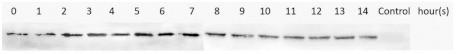

Footnote: Due to the limited well numbers, the samples were separated on two polyacrylamide gels and then transferred onto two NC membranes. To better visualize the protein expression in this study, the two blots were combined. Lanes 7 and 8 were seemingly slightly incoherent upon the combination of blots.

## Data Availability

The data presented in this study are available on request from the corresponding author.

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
