# Peer review of "Expression of Influenza M2e-NP Recombinant Fusion Protein in Escherichia coli BL21 (DE3) and Its Binding to Antibodies"

_vaccines, 2022, doi:10.3390/vaccines10122066_

Round 1
Reviewer 1 Report
Synthesis of Influenza M2e-NP Recombinant Fusion Protein in 2 Escherichia coli BL21 (DE3) and Its Antigenicity
The authors explained the effectiveness of M2e-NP as a universal vaccine candidate against the continuously emerging influenza A virus. I was able to understand the explanation easily, but I would like to ask you a few questions about the manuscript.
1. In this study, M2e-NPs were formed into VLPs, showing potential as vaccine formulations. Although the results of promoting effective immunity as a vaccine (high antigenicity and immunogenicity) are sufficient, it would be nice to have additional results on the biochemical, biophysical, and biological properties that are essential components of VLP-based vaccines.
2. I would appreciate it if you could add the pRSET B-M2e-NP expression plasmid information used in this experiment to the figure through the scheme.
3. According to table 4, protein induction was performed using 0.5 mM IPTG at 37°C for the Optimization of M2e-NP and mM2e-NP protein expression. Western blot results show that in IPTG condition experiments conducted at 37 °C, quantitative data would be desirable to show the difference between the two data, 0.5 mM and 1 mM.
4. In this paper, including introduction and 3.3, six-point mutations that lead to changes in amino acids in proteins were detected in the M2e-NP fusion gene. However, according to 2.3, DNA sequencing and BLAST results indicate seven-point mutations that caused amino acid changes in the M2e-NP fusion gene, one in the M2e gene and the other six in the NP gene. I want to know the detailed explanation of this.
5. You need to redraw the graph with another program (Sigma, Prism).
6. Expressions of M2e-NP and mM2e-NP were well compared at 25℃ and 37℃. Additionally, the experimenter's discussion of why the expression of mM2e-NP is higher than that of M2e-NP at 25℃ should be added. In addition, the standard was set at 37℃, 0.5mM IPTG, and 5 hours for optimal production of M2e-NP and mM2e-NP, and the rationale for setting the time to 5 hours should be added. In the ELISA data, the error bar of the data was displayed well. However, it is necessary to additionally mention the number of repetitions of the experiment.
7. To test the sensitivity of the antibody to mM2e-NP and M2e-NP, 50-1500 ng of protein was coated on wells. It is said that at least 500 ng of protein is required for antibody binding for the recommended titer, but to support this, it is necessary to measure the standard of proteins with significantly different OD values between 100 and 500 ng.
8. In Result 3.1, the M2e gene 211 (132 bp), on the other hand, was only amplified at 71℃ (Figure 1, right panel, lane 5). 212 Primer-dimers were observed at the bottom of the gel when the reaction was performed Unlike the sentence in 213 at 55-67.2°C., in Result 3.2, there is an annealing result at 55°C, so additional is required.
9. Western blotting is an analysis method for qualitative analysis of proteins, but in the results of this study, the background of the connection part of 7 and 8 is not uniform and the band is also uneven. The authors need to redraw the results.
Author Response
- In this study, M2e-NPs were formed into VLPs, showing potential as vaccine formulations. Although the results of promoting effective immunity as a vaccine (high antigenicity and immunogenicity) are sufficient, it would be nice to have additional results on the biochemical, biophysical, and biological properties that are essential components of VLP-based vaccines.
Response: Due to the lack of funding for additional experiments, the authors are unable to carry out more experiments. However, more details on VLPs as ideal carriers for vaccines are provided (lines 403, 406-408).
- I would appreciate it if you could add the pRSET B-M2e-NP expression plasmid information used in this experiment to the figure through the scheme.
Response: The gene arrangement scheme of the expression plasmid was added in the manuscript. (lines 128-133).
- According to table 4, protein induction was performed using 0.5 mM IPTG at 37°C for the Optimization of M2e-NP and mM2e-NP protein expression. Western blot results show that in IPTG condition experiments conducted at 37 °C, quantitative data would be desirable to show the difference between the two data, 0.5 mM and 1 mM.
Response: A quantitative comparison of the expression yields is now added in the revised manuscript (lines 281-284, 308-310).
- In this paper, including introduction and 3.3, six-point mutations that lead to changes in amino acids in proteins were detected in the M2e-NP fusion gene. However, according to 2.3, DNA sequencing and BLAST results indicate seven-point mutations that caused amino acid changes in the M2e-NP fusion gene, one in the M2e gene and the other six in the NP gene. I want to know the detailed explanation of this.
Response: There were typos in section 2.3. It was supposed to be six point mutations in total, in which one was found in the M2e whereas the other five were located in the NP. An amendment is now made to section 2.3 (lines 135 & 137).
- You need to redraw the graph with another program (Sigma, Prism).
Response: All the graphs have now been re-plotted using the GraphPad Prism version 9 (Figures 4 and 7).
- Expressions of M2e-NP and mM2e-NP were well compared at 25℃ and 37℃. Additionally, the experimenter's discussion of why the expression of mM2e-NP is higher than that of M2e-NP at 25℃ should be added. In addition, the standard was set at 37℃, 0.5mM IPTG, and 5 hours for optimal production of M2e-NP and mM2e-NP, and the rationale for setting the time to 5 hours should be added. In the ELISA data, the error bar of the data was displayed well. However, it is necessary to additionally mention the number of repetitions of the experiment.
Response: A discussion point is now added to explain the higher expression of mM2e-NP at 25°C (lines 450-453). The rationale of choosing five hours as the expression duration is now added (lines 446-448). The number of replicate (n=3) is now mentioned in the captions (lines 383-387).
- To test the sensitivity of the antibody to mM2e-NP and M2e-NP, 50-1500 ng of protein was coated on wells. It is said that at least 500 ng of protein is required for antibody binding for the recommended titer, but to support this, it is necessary to measure the standard of proteins with significantly different OD values between 100 and 500 ng.
Response: The remarkable difference between 100 and 500 ng of proteins is now represented by a p-value in the graph caption and text (lines 365, 383-387).
- In Result 3.1, the M2e gene 211 (132 bp), on the other hand, was only amplified at 71℃ (Figure 1, right panel, lane 5). 212 Primer-dimers were observed at the bottom of the gel when the reaction was performed Unlike the sentence in 213 at 55-67.2°C., in Result 3.2, there is an annealing result at 55°C, so additional is required.
Response: The annealing temperature 71℃ was the specific annealing temperature for amplifying the M2e gene, meanwhile the 55℃ mentioned in section 3.2 was used to fuse the genes, hence they were two different experiments. To prevent confusion, a sentence to indicate the use of 55°C is now added to the text (lines 236-237).
- Western blotting is an analysis method for qualitative analysis of proteins, but in the results of this study, the background of the connection part of 7 and 8 is not uniform and the band is also uneven. The authors need to redraw the results.
Response: In this study, in order to better present the results, a total of 15 samples were run separately using wider wells (n=8 per gel) on two polyacrylamide gels. As a result, the protein transfer also required two NC membranes which then generated two western blots. To prevent biases in the data interpretation, the two blots were processed and developed simultaneously, and presented side-by-side in this study. Redraw the blots might trigger doubts in readers and also violate the research ethics. We therefore remain the blots, and add a footnote underneath Table 4 to briefly explain the incoherence in the blots (lines 304-306).
Reviewer 2 Report
Brief summary
In this study authors have produced a recombinant fusion protein from influenza virus; M2e-NP and a mutant of it. They expressed and purified from E coli; and validated it by ELISA and western blot assay. Though, manuscript is well written, however, justifying following comments will be further strengthen the manuscript –
Comments-
1. Tittle needs to be modified for a clear message. The words synthesis and antigenicity are ambiguous. As, synthesis mostly referred for chemical synthesis, and antigenicity mostly refer as strength of immune response after immunisations.
2. Sentence in line 20-21 “The recombinant plasmid (pRSET B-M2e-NP) was 20 introduced into Escherichia coli (E. coli) BL21 (DE3) for protein expression” can be moved to somewhere after line 23, to keep order of experiments- cloning followed by expression.
3. Sentence in line 88-89 “In conclusion, M2e-NP is a promising universal influenza vaccine candidate that can promote effective immunity against influenza viruses” can be modified to the conclusion of the study, not for the possibilities. Here there is no experiment to support its vaccination potency.
4. Since in figure 5 OD values were plotted, it should be clearly mentioned whether those plotted OD values were raw values or normalised/subtracted with blank controls (it can be mentioned in material and methods). As in figure5(i) it looks all blank controls (0 ng) have OD=0, so it is less likely they are raw ODs.
5. In ELISA plot, mention what is the n value (replicates) for shown errors.
6. Citation 22 does not seem to support statement. Cite original research article.
7. Line 404-405, how cited references are relevant for this statement, justify or modify. Same for reference 35 in line 438.
8. Line 445-448, if epitope of tested antibody is known, It can be discussed which amino acids of NP could be possibly causing this differential effect. Since those five mutations are spanning in >100aa, it is less likely all those residues involved in the epitope of that tested antibody.
9. Line 454-456, “This implies that the antibody has high avidity to the fusion protein, and thereby suggests the high immunization efficacy of the fusion protein” this statement, does not seem to be justified need to modify.
10. Written conclusion line 460-470, is mostly extrapolations and nothing mentioned about the conclusion of this study. Need to re-write.
Author Response
Reviewer 2
Comments
- Tittle needs to be modified for a clear message. The words synthesis and antigenicity are ambiguous. As, synthesis mostly referred for chemical synthesis, and antigenicity mostly refer as strength of immune response after immunisations.
Response: The title is now modified as requested (lines 1-3).
- Sentence in line 20-21 “The recombinant plasmid (pRSET B-M2e-NP) was 20 introduced into Escherichia coli (E. coli) BL21 (DE3) for protein expression” can be moved to somewhere after line 23, to keep order of experiments- cloning followed by expression.
Response: This part has been amended as suggested (lines 22-24).
- Sentence in line 88-89 “In conclusion, M2e-NP is a promising universal influenza vaccine candidate that can promote effective immunity against influenza viruses” can be modified to the conclusion of the study, not for the possibilities. Here there is no experiment to support its vaccination potency.
Response: In order to prevent confusion, this line is now deleted from the Introduction.
- Since in figure 5 OD values were plotted, it should be clearly mentioned whether those plotted OD values were raw values or normalised/subtracted with blank controls (it can be mentioned in material and methods). As in figure5(i) it looks all blank controls (0 ng) have OD=0, so it is less likely they are raw ODs.
Response: The OD readings were normalized to that of the blank controls prior to presenting in the graph. Normalization of the readings is now mentioned in section 2.7 (lines 214-215).
- In ELISA plot, mention what is the n value (replicates) for shown errors.
Response: The number of replicate is now added in the caption (lines 383-387).
- Citation 22 does not seem to support statement. Cite original research article.
Response: The citation is now changed and replaced by a more relevant study.
- Line 404-405, how cited references are relevant for this statement, justify or modify. Same for reference 35 in line 438.
Response: Those citations are now changed and replaced by more relevant ones.
- Line 445-448, if epitope of tested antibody is known, It can be discussed which amino acids of NP could be possibly causing this differential effect. Since those five mutations are spanning in >100aa, it is less likely all those residues involved in the epitope of that tested antibody.
Response: The rationale of the discussion is now added (lines 467-470).
- Line 454-456, “This implies that the antibody has high avidity to the fusion protein, and thereby suggests the high immunization efficacy of the fusion protein” this statement, does not seem to be justified need to modify.
Response: The line is now modified to “This implies that the antibody has a reasonably good binding to the fusion protein” (lines 485-486).
- Written conclusion line 460-470, is mostly extrapolations and nothing mentioned about the conclusion of this study. Need to re-write.
Response: The conclusion is now amended (lines 491-503).